# Investigation of the Role of Nano-Titanium on Corrosion and Thermal Performance of Structural Concrete with Macro-Encapsulated PCM

**DOI:** 10.3390/molecules24071360

**Published:** 2019-04-06

**Authors:** Ehsan Mohseni, Waiching Tang, Shanyong Wang

**Affiliations:** 1School of Architecture and Built Environment, the University of Newcastle, Callaghan, NSW 2308, Australia; 2ARC Centre of Excellence for Geotechnical Science and Engineering, the University of Newcastle, Callaghan, NSW 2308, Australia; Shanyong.wang@newcastle.edu.au

**Keywords:** phase change materials, thermal energy storage aggregates, nano-titanium, accelerated corrosion test, thermal conductivity

## Abstract

The present study aims to investigate the impact of thermal energy storage aggregate (TESA) and nano-titanium (NT) on properties of structural concrete. TESA was made of scoria encapsulated with phase change materials (PCMs). Coarse aggregates were replaced by TESA at 100% by volume of aggregate and NT was added at 5% by weight of cement. Compressive strength, probability of corrosion, thermal performance, and microstructure properties were studied. The results indicated that the presence of TESA reduced the compressive strength of concrete, although the strength was still high enough to be used as structural concrete. The use of TESA significantly improved the thermal performance of concrete, and slightly improved the resistance of corrosion in concrete. The thermal test results showed that TESA concrete reduces the peak temperature by 2 °C compared to the control. The addition of NT changed the microstructure of concrete, which resulted in higher compressive strength. Additionally, the use of NT further enhanced the thermal performance of TESA concrete by reducing the probability of corrosion remarkably. These results confirmed the crucial role of NT in improving the permeability and the thermal conductivity of mixtures containing PCM. In other words, the charging and discharging of TESA was enhanced with the presence of NT in the mixture.

## 1. Introduction

Management of energy consumption and determination of energy reduction strategies in buildings are associated with the assessment of the types of energy required for the building (heating, cooling, and electrical). In other words, energy consumption management presents a method to measure the actual energy requirements of a building, and it also proposes appropriate solutions to reduce and minimize the consumption. Today, the use of contemporary methods and renewable energies for building consumption management receives significant attention worldwide. Global population growth, reduced energy sources, and increased fossil fuel emissions have resulted in global communities using renewable energies all over the world. Among the renewable energies, the sun is an endless source of energy. Given that the intensity of sunlight varies throughout the day, the use of this energy without proper storage is not feasible. Therefore, the issue of solar energy storage is very important and has attracted many researchers over the last few years. One of the most important ways to use solar energy is the storage of solar energy during hot hours and release of the energy when the temperature is lower. When materials with a high thermal capacity, such as rock, concrete, clay, brick, and water are used as primary building materials, significant amounts of sunlight are absorbed and stored during the day, and this energy is released at night to regulate the indoor temperature. However, this traditional method has some disadvantages, including the extensive use of building materials and increasing the dead load on the building. One of the new and promising methods of solar energy storage is the use of materials with high energy storage. 

One of the most effective materials for storing thermal energy is phase change materials (PCMs). Their performance is such that, when the temperature is increasing, the heat energy is absorbed and will be released by decreasing the temperature. One application of these materials is as insulation in building walls. Thermal energy is stored in two ways, sensible and latent heat. The energy stored by latent heat associated with the phase change of the material is more important because of the high density of thermal energy storage. Although all materials have the ability to change the phase, it is worth noting that the substance known as PCM can change phase over the performance temperature range of the system. For example, when the solid phase changes to liquid, the material stores the thermal energy, and when necessary, releases the energy by freezing. The phase change operation can take place quickly or be delayed depending on the PCM type.

Organic PCMs, including paraffin and nonparaffin, render congruent melting without phase segregation and supercooling throughout the large number of melting/freezing cycles at the cost of latent heat of fusion. They undergo solid-liquid phase transition during heating and subsequent cooling. Several organic PCMs have been discovered and categorized over the years [1,2]. Despite several benefits of organic PCMs, the main drawbacks to overcome, include their low thermal conductivity which results in a low charging/discharging rate, and their flammability and leakage in containers [3,4].

PCMs can be incorporated into the system by the encapsulation method which can be further classified as micro- and macro-encapsulation methods. Extensive use of micro- and nano-encapsulated PCMs has a higher cost and an obvious adverse effect on either the mechanical strength or thermal conductivity of the resulting building materials [5,6,7]. In contrast, macro-encapsulated PCM stored into comparatively larger carriers (e.g. tubes, spheres, pouches, panels, spheres, and lightweight aggregates) can be added to building elements with little influence on the structure and function of these elements in the buildings [8,9,10,11,12]. In the macro-encapsulation technique, macroscopic containers (encapsulation size from a few millimeters up to several liters) such as tubes, poaches, spheres, and panels carry PCMs for subsequent use in construction elements [13]. Macro-encapsulation improves the compatibility of PCM with the surrounding matrix by acting as a barrier and preventing the leakage of liquid PCM. Since the containers usually have the capability to accommodate the internal volume change of PCM, the external volume changes are minimized. The main problem with this application is the poor thermal conductivity of the PCM/container which results in a substantial reduction in the heat transfer performance of the PCM. One of the effective ways to overcome the poor conductivity of PCMs is the use of metal or carbon-based nanoparticles.

Nanoscale materials are taken into consideration by industry and academia in light of their significant behaviors [14,15,16,17]. Meanwhile, the concrete industry is one of the important users of nanostructured materials in terms of strength and durability, according to its needs [18,19,20]. The use of nanoscale materials in the building industry has resulted in new structures that no longer depend on natural resources in their construction. This has been done by modifying structures on a nanoscale, or by applying different structures and promoting reactions. Subsequently, different properties including permeability, magnetic properties, microstructure compaction, and electrical conductivity have been improved [21].

Although many studies have been conducted using PCMs in non-structural applications, studies on thermal conductivity, corrosion, and leakage problems still need to be investigated for structural concretes. This study aims to address this gap by evaluating the compressive strength, thermal performance, and corrosion of thermal energy storage (TES) concretes made by macro-encapsulated PCM and the application of nano-titanium (NT).

## 2. Results and Discussion

### 2.1. Compressive Strength

The compressive strength test results of samples at the ages of 7, 28, and 90 days are shown in Figure 1. From the results, it is clear that the strength increased with time. The compressive strength of thermal energy storage aggregate (TESA) concrete samples showed lower values at all ages as compared with the control sample. Concrete containing 100% TESA showed an average reduction, of 27%, in compressive strength. This reduction in compressive strength when using TESA was consistent with other studies [8,10,22]. The explanations for this are as follows: First, the lower strength and stiffness of TESA as compared with natural coarse aggregates tend to fail and crack under loading. Second, the damaged TESA leads to stress concentration during loading. It is worth noting that the compressive strength values were still high enough to be used as structural concrete. The local standard requires a 28-day strength to be greater than 20 MPa, however, a strength of approximately 35 MPa was achieved in this study.

The addition of NT to the mixtures increased the compressive strength by approximately 25%. The increase in strength confirms that nanoparticles have the ability to fill the pores and make the sample compact. The compressive strength achieved at 28 days was approximately 45 MPa in the TESA-NT mixture. The microstructure of samples with SEM images is described in Section 2.4.

### 2.2. Corrosion Test

Figure 2 displays a contour plot corresponding to the potential corrosion within the sample at each reference point across the face of the sample. The results for half-cell potential were then compared to the range for potential corrosion within the sample. According to ASTM C876-09 [23], the lower the reading in millivolts the higher the probability for corrosion within the sample. The referenced values for half-cell potential and its corresponding corrosion probability are included in Table 1. 

As shown in Figure 2a, the possibility of corrosion in the control sample was more than 90% on the left side of the sample, and approximately 50% on the right side. This shows that the sample was vulnerable to corrosion which indicates its low durability. However, the replacement of TESA did not have a negative impact on the probability of corrosion. As shown in Figure 2b, the possibility of corrosion seemed to be reduced as compare with the control. From these results, it is concluded that PCM was properly encapsulated in the lightweight aggregate (LWA) and the coverage prevented the leakage of PCM. It is worth noting that the addition of NT showed a significant influence on the possibility of corrosion. As is shown in Figure 2c, the red area, which represented the high probability of corrosion, was reduced noticeably. The possibility of corrosion at the center was less than 50% and it was reduced to approximately 10% on the right-hand side. 

Figure 3 shows the average and maximum corrosion values recorded on the surface of the samples. It is clear that the replacement of TESA did not have a detrimental effect on the corrosion of the sample and the addition of NT improved the sample’s resistance against corrosion. Probably, the main reason is a function of NT which increases the durability by filling the pores in micro- and nanoscales and thus reduces the possibility of moving ions within the matrix. However, the specimens generally showed similar rusting and cracking. 

Within a concrete structure, steel reinforcement plays a crucial role as the stem of the tensile strength within the structure. While considering this, it is important to understand the rate of corrosion that may affect the serviceability performance of the structure. An accurate way to understand the corrosion rate of steel is to observe the overall loss of mass as a percentage of the rebar’s original weight. The results recorded from this experiment are presented in Figure 4. A control rebar was used and submerged in the solution with no concrete covering. A current was sent through this single piece of rebar in an identical fashion to the other samples and the corrosion conditions were observed and recorded.

The results are similar to those observed in the accelerated corrosion test which show a decrease in corrosion with the presence of TESA and NT. It was found that the use of TESA and TESA-NT reduced the mass loss by 9% and 56%, respectively, as compared with the control mixture. 

### 2.3. Thermal Performance

The thermal performance of concrete panels made of normal aggregate, TESA, and TESA-NT was evaluated by measuring the variations of temperature at different positions on concrete panels, i.e., at the center of the test room, and interior and exterior surfaces of the panel. Previous thermal studies [22] on coated LWA and TESA indicated that TESA showed a higher thermal performance as compared with coated LWA. This study focused on the influence of TESA and NT on the thermal performance of concrete. The chamber temperature was heated for 4 hours and then cooled down. All the samples experienced the same conditions in order to have an accurate comparison of their thermal performance. The measurement results of temperature variations at different positions are shown in Figure 5.

In general, similar temperature variation trends were observed at different positions of the test room and the concrete panels. The test room of the control sample underwent a remarkable increase in temperature during the heating period and the temperature decreased rapidly during the cooling period. In the case of concrete containing TESA, the rate of temperature increase decreased which resulted due to the heat absorption ability of PCM during the heating period. The peak temperature of the sample with TESA was less than the control sample by approximately 2 °C. Later, during the cooling process, the release of heat was slower in the TESA sample as compared with the control sample, which is also due to the heat release ability of macro-encapsulated PCM. 

When NT was used in conjunction with TESA, an obvious temperature difference was found as compared with the control and the TESA-NT samples. This substantiates the effect of nanoparticles on thermal performance, and this is attributed to their higher thermal conductivity. Therefore, it can be concluded that the addition of TESA-NT increases the thermal conductivity of concrete because of high conductive particles in the matrix. As a result, the charging and discharging of the TESA was enhanced remarkably.

It is expected that if macro-encapsulated PCMs with NT are used in the structural or non-structural members of buildings located in moderate climate zones, the thermal comfort and fluctuation of temperature inside the buildings will be significantly improved.

### 2.4. Microstructure Properties

It is known that interfacial transition zone (ITZ) between different materials used in the matrix plays a critical role in mechanical properties. The ITZ occupies a low ratio of the matrix, however, it has significant influences on the physical and mechanical properties. Well compacted ITZ is better, since higher load transfer and compressive strength is achieved. 

To observe and understand the effectiveness of a coating system on TESA, the microstructure of concrete with TESA and concrete with LWA were studied using SEM as shown in Figure 6. As shown in Figure 6b, the ITZ between TESA-cement paste is compacted, and there are no cracks in the coating materials.

Although the use of TESA reduces the compressive strength, the high quality of ITZ between the LWA and the coating materials and the ITZ between the coating materials and the cement paste contributes to the effective transfer of loads, and therefore the strength is not drastically reduced. Additionally, the high-quality of the coating materials prevents leakage of the PCM. 

The microstructure of the samples with NT was evaluated using SEM and EDS analyses and the results are illustrated in Figure 7. The results show that there is a dense microstructure in concrete containing NT. Additionally, no large pores were observed in the SEM images and the homogeneous paste show the filling ability of the NT particles.

If a proper content of nanoparticles is well distributed, the crystallization will be controlled through restricting the growth of Ca(OH)_2_ crystals by the nanoparticles [24,25]. In fact, by using nanoparticles, the content of Ca(OH)_2_ crystals is reduced and the large pores are filled. In addition, NT enhances the density of the microstructure as well as the compressive strength of the concrete. This is due to the rapid formation of C-S-H gel which accelerates the hydration of the cement. On the whole, nanoparticles significantly decrease the volume of pores.

## 3. Experimental Program

### 3.1. Materials

The scoria supplied from Gardenworld company (Newcastle, Australian) was used as the porous lightweight aggregate (LWA) with an average particle size of 13 mm. The density, porosity, and water absorption of scoria were 1.581 g/cm^3^, 56.55%, and 16%, respectively. The highly porous nature of LWA was illustrated in the scanning electron microscopy (SEM) (ZEISS SIGMA VP, Germany) image (Figure 8) as well as the Mercury intrusion porosimetry (MIP) results as shown in Figure 9 and Table 2. The nano-titanium was supplied from US Research Nanomaterials, Inc. (Houston, TX, USA) having a nominal diameter of 30 nm, purity of 99.98%, and specific surface area of 50 m^2^/g and they were used to improve the properties of the TESA concretes. Figure 10 shows the transmission electron microscopy (TEM) (JEOL, Tokyo, Japan) image of NT used in this study. PureTemp 23 supplied from PureTemp^®^ (Plymouth, MN, USA) was used as an organic commercial grade PCM with a specified melting temperature of 23 °C. The PCM thermal properties were evaluated via a differential scanning calorimeter (DSC) by heating the samples from −10 °C to 50 °C with a linear heating rate of 5 °C/min under nitrogen atmosphere at a flow rate of 50 ml/min (Figure 11). The PCM thermal stability was assessed using the thermogravimetry analysis (TGA) (Perkin-Elmer Diamond TGA, Melbourne, Australia) by heating the PCM samples from 25 °C to 600 °C with a linear heating rate of 10 °C/min under inert gas flow (Figure 12). Type II Portland cement and fly ash in accordance with AS 3972 and AS 3582.1, respectively, were used as cementitious materials. River sand with a density of 2600 kg/m^3^ and maximum particle size of 4.75 mm, and granite gravel with a density of 2600 kg/m^3^ and an average particle size of 10 mm were used as normal weight aggregates. Figure 13 shows the details of the size distribution of aggregates used in this study. To prepare the thermal energy storage aggregate (TESA), epoxy resin adhesive and hardener supplied from Sikadur^®^ (Sika, Australia) in accordance with ASTM C881-78 were applied to coat the surface of LWA encapulsated with PCM. Furthermore, silica fume was used to separate and cover the epoxy coated aggregates. The scoria aggregates, before and after coating, are shown in Figure 14. A high range water reducer (HRWR) was used to disperse the nanoparticles and to attain the desired workability of the concrete mixtures.

### 3.2. Mix Proportions and Fabrication of TESA

Three mixes were designed to characterize mechanical, thermal, and corrosion properties of the TESA concretes. Details of the proportions of mixes are given in Table 3. The control sample was made of normal aggregates without PCM. In the second mixture, 100% by volume of the normal coarse aggregate was replaced by the TESA. In the third mixture, 100% by volume of normal coarse aggregate was replaced by the TESA with an addition of 5% NT (TESA-NT).

The cross sections of the control and the TESA sample are illustrated in Figure 15. The water to cement ratio was 0.35 and it was constant for all mixes.

The vacuum impregnation technique was used to promote the rapid expulsion of air entrapped in the LWA and allowed absorption of a large quantity of liquid PCM. First, the LWA and the melted PCM were contained in a bowl placed inside the vacuum chamber for 30 min at a pressure of 0.1 MPa until all visible air bubbles were swept. Then, the LWA-PCM samples were transferred to a refrigerator at 4 °C to keep the PCM solidified. During the vacuum process, the temperature of the test setup was kept above the melting point of PCM. Later, the LWA-PCM was immersed for 10 min in the mixture of epoxy. To prepare the epoxy mixture, the adhesive and the hardener were mixed at a constant ratio of 2:1.2. Afterwards, the aggregates coated with epoxy were distributed in a tray filled with silica fume to separate the aggregates. Then, the aggregates were put into the baskets for 7 days to set the epoxy. 

The mixing procedure was as follows: Natural coarse aggregates (if applicable), fine aggregates, and cement and fly ash were dry mixed at a moderate mixing speed (80 rpm) for 1 min to homogenize the dry components. Then, 70% of mixing water and a mixture of NT were added and mixed for 90 s. Next, the remaining water and mixture of NT were added and mixed at a high speed (120 rpm) for a further 30 s. The mixture was allowed to rest for 90 s, and finally mixing was continued for an additional 1 min. It is important to note, TESA must be added as the last component for the mixing of the mixtures containing the TESA in order to avoid possible damage to the macro-encapsulated TESA during the mixing process.

It is likely that the NT particles may not be uniformly distributed in the mixture because they tend to agglomerate due to their large surface area and energy [19]. To achieve a uniform dispersion, the nanoparticles and HRWR were mechanically stirred for 15 min at 600 rpm. The mixture then underwent a procedure using an ultrasonic probe sonicator which was applied for 90 min. The temperature of the mixture increased during the sonication process, therefore, to reduce the temperature of the sample the bowl was placed in a bath filled with ice, and the temperature of the mixture was monitored every 15 min.

### 3.3. Testing Methods

The compressive strength was determined for cylinders with a diameter of 100 mm and a length of 200 mm according to AS1012.9 [26] after 7, 28, and 90 days of curing.

To assess the corrosion risk of reinforcement in concrete specimens, the most common methods are linear polarization resistance (LPR) [27] and electrochemical impedance spectroscopy (EIS) [28]. Recently, the accelerated corrosion test (ACT) has been widely used among researchers to induce corrosion of steel embedded in concrete for a relatively short period [29,30,31]. They measured half-cell potential readings using a half-cell potentiometer (HCP) positioned at different points on the concrete prisms to quantify the high impedance potentials in the concrete, which indicated the probability of corrosion. In this study, the probability of corrosion was studied using the half-cell potential test in accordance with ASTM C876-15 [23] by measuring the potential difference between the embedded steel bar and a reference electrode. In order to complete the tests, two prismatic samples were cast for each of the concrete mixes. The samples consisted of a single steel reinforcement bar with a length of 290 mm and a diameter of 12 mm that was weighed prior to casting.

Corrosion tanks were constructed and filled with a solution containing water with 3.5% salt (NaCl) to assist in accelerating the corrosion within the sample. When samples reached their 28-day curing strength they were placed in their respective corrosion tanks and connected to an electrical current. The electrical schematic included running a current through the steel reinforcement bar of each sample, which becomes the anodic member of the system. Whereas, a stainless-steel sample is included in each corrosion schematic to act as the cathodic member within the galvanic cell. The electrical schematic is shown in Figure 16.

After ACT for 90 days, the potential corrosion of the samples was measured using the half-cell potential method, as shown in Figure 17. A grid pattern was glued to the surface of each sample to provide a point of reference for measuring potential using a potassium chloride probe. The sample was installed into an electrical circuit with a voltmeter, and when the probe was applied to the surface of the sample, the voltage was measured and recorded. In order to record a constant level of potential within the sample, the time applied to each point of reference remained constant at 20 seconds. Since the voltmeter was calibrated to record values corresponding to copper sulphate, whereas our voltmeter was calibrated corresponding to potassium chloride, our readings were corrected by a value of −0.06 in order to accurately map the surface potential of each of the samples. The equation for correction is outlined in Equation (1):
Corrected value (mV) = 1000 × (recorded value (V) − 0.06).
(1)

After the ACT test and before weighing, to evaluate the total steel mass loss (i.e. corrosion) of each steel reinforcement, the samples were carefully broken open and the rebar was removed and cleaned of any corrosion using acid. The percentage of rebar mass loss due to corrosion was recorded as a percentage of its original weight.

A self-designed heating and cooling system was employed to study the thermal performance of the TESA concrete as shown in Figure 18. The setup consisted of a small test room, a split air conditioner system (as a heating/cooling source), polystyrene envelope, and a wooden box (1 m × 1 m × 1 m) with an opening. Thermocouples (Type K resolution ±0.3 °C) were placed in the center of the test room and at the inner and outer surfaces of the concrete panel. The readings were recorded using a data-logger platform and displayed on the computer. The test room model consisted of six panels, five of which were made up of wood and enclosed in 40 mm of polystyrene foam, while the top panel was covered with the TESA concrete. The dimensions of the top panel were 200 × 200 × 40 mm^3^. The complete thermal test set-up is illustrated in Figure 18.

Moreover, the ITZ between cementitious materials and coating materials, the microstructure and the porosity of samples were assessed using the SEM.

## 4. Conclusions

Replacement of natural aggregates by TESA reduced compressive strength at all ages. Although, concrete containing TESA exhibited a 27% reduction of compressive strength on average, the compressive strength values were still high enough to be used as structural concrete.

The addition of NT to the mixtures increased the compressive strength by approximately 25%, which confirmed the ability of nanoparticles to fill the pores and to make the sample compact. The compressive strength achieved at 28 days was about 45 MPa in the TESA-NT mixture.

The possibility of corrosion in the control sample was more than 90% on the left side of the sample, and approximately 50% on the right side. The results indicated that the sample is vulnerable to possible corrosion in an aggressive environment.

The replacement of TESA did not have a detrimental impact on the probability of corrosion. The results showed that the PCM was properly encapsulated in the LWA and the coating prevented leakage of the PCM.

The addition of NT has a significant influence on the corrosion possibility. The corrosion possibility at the center was less than 50% and on the right-hand side it was approximately 10%. The results showed that the replacement of TESA and TESA-NT reduced the rebar mass loss by 9% and 56%, respectively, as compared with the control mixture.

By replacing PCM, the rate of temperature increase decreased which is due to the ability of PCM in heat adsorption during the heating period. The peak temperature of the sample with TESA was less than the control sample by approximately 2 °C.

When NT was used in conjunction with TESA, an obvious temperature difference was found compared to the control and the TESA-NT samples. This substantiates the effect of nanoparticles in thermal performance which is attributed to their higher thermal conductivity. As well, the charging and discharging of TESA was enhanced remarkably. 

The results showed that thermal comfort and fluctuation of the temperature inside a building can be improved by using macro-encapsulated PCM with NT in the structural or non-structural members of buildings in moderate climate zones.

The SEM images indicated that the ITZ between the LWA coating system and the cement paste was compacted, and no cracks were seen in the coating materials. No large pores were observed in the SEM images of the homogeneous paste which indicated the filling ability of the NT particles.

## Figures and Tables

**Figure 1 molecules-24-01360-f001:**
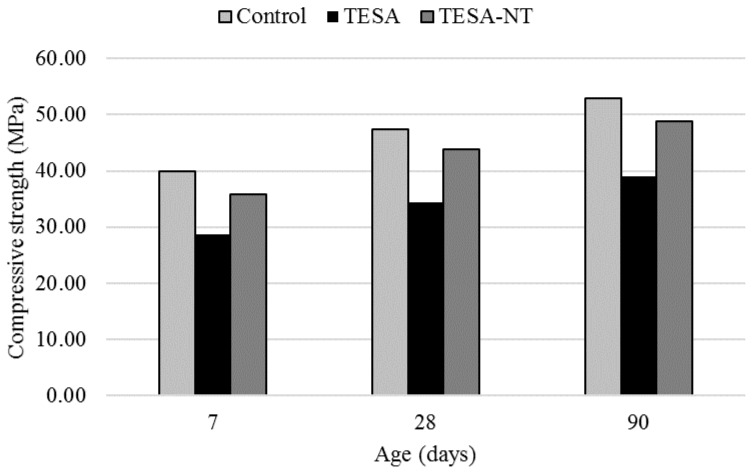
Compressive strength test results of samples at 7, 28, and 90 days of curing.

**Figure 2 molecules-24-01360-f002:**
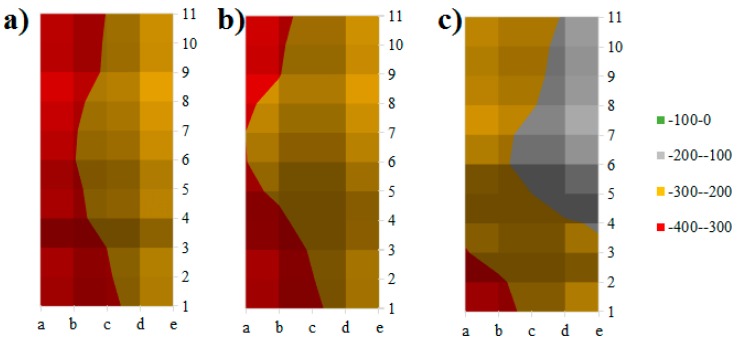
Mapping corrosion potential across the surface of a sample, (**a**) control, (**b**) thermal energy storage aggregate (TESA), and (**c**) TESA-nano-titanium (NT).

**Figure 3 molecules-24-01360-f003:**
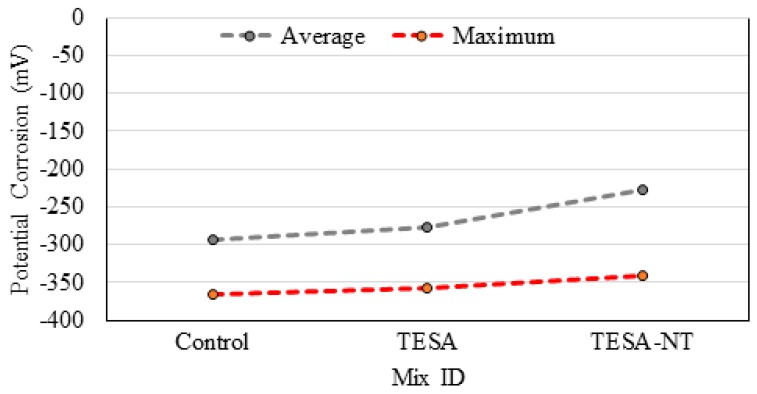
Corrosion potential after accelerated corrosion.

**Figure 4 molecules-24-01360-f004:**
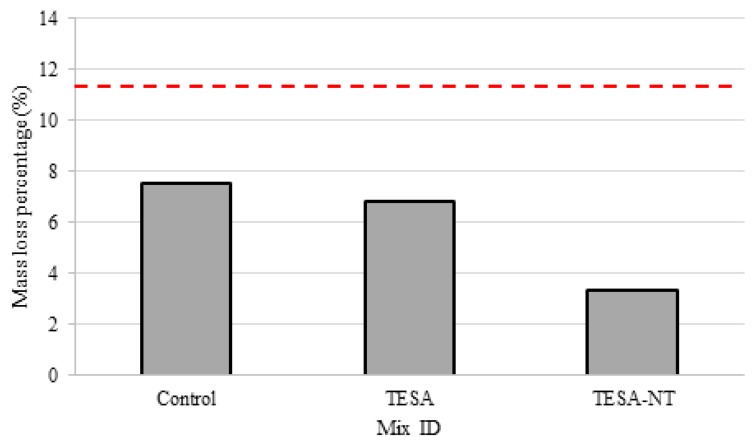
Percentage of rebar mass loss during accelerated corrosion test.

**Figure 5 molecules-24-01360-f005:**
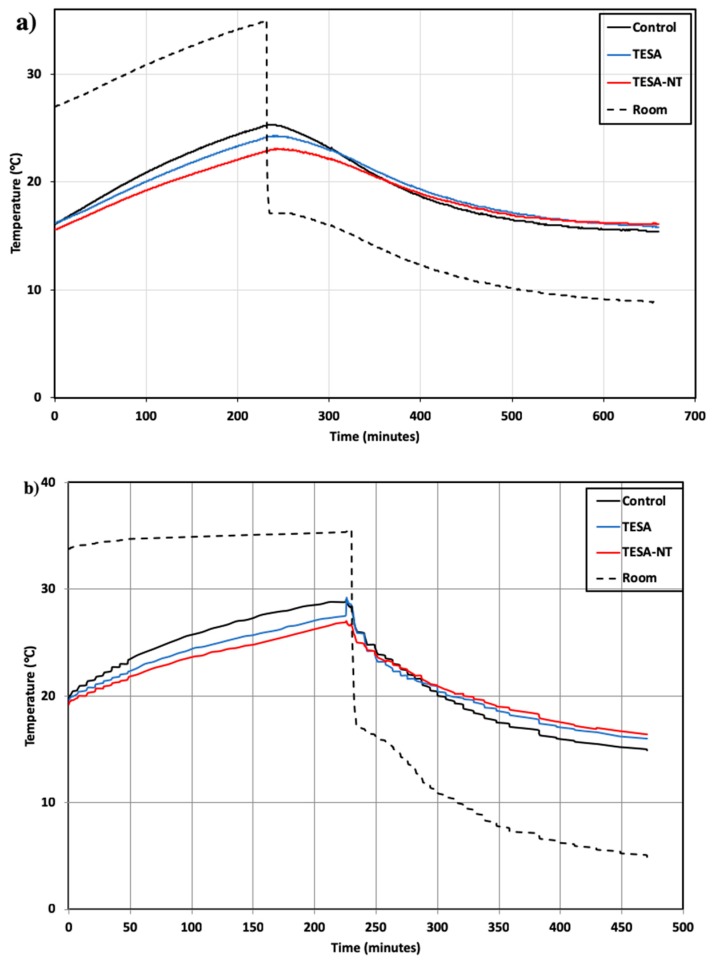
Thermal performance test results, (**a**) in the center of the test room, (**b**) at the inner and (**c**) outer surfaces of the concrete panel.

**Figure 6 molecules-24-01360-f006:**
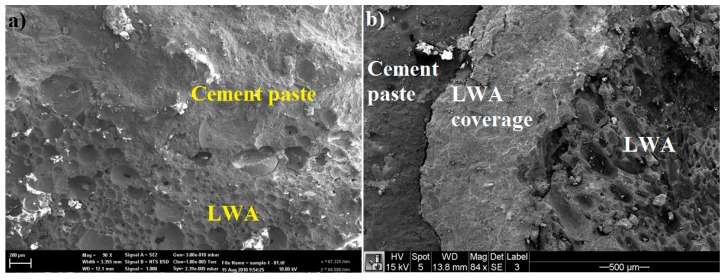
SEM images of samples: (**a**) without coating materials and (**b**) with lightweight aggregate (LWA) coverage.

**Figure 7 molecules-24-01360-f007:**
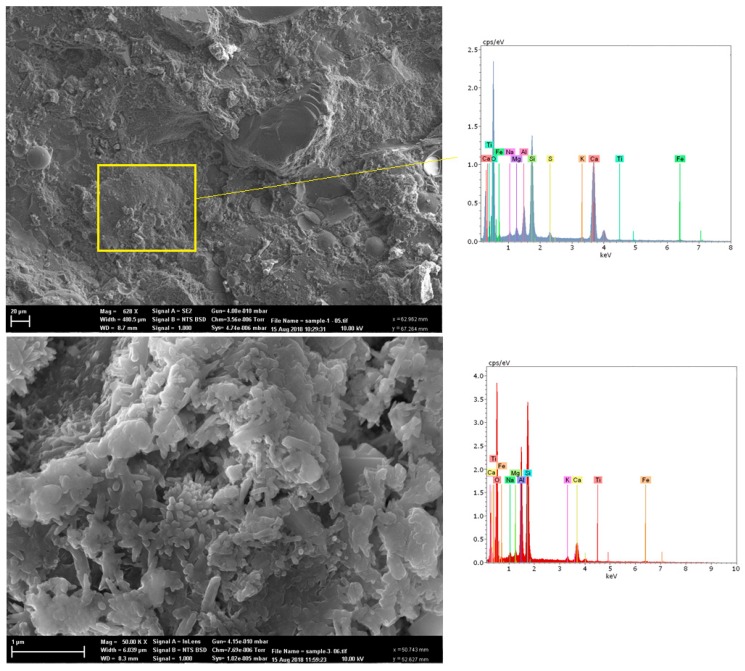
SEM and EDS images of sample containing nano-titanium at different magnifications.

**Figure 8 molecules-24-01360-f008:**
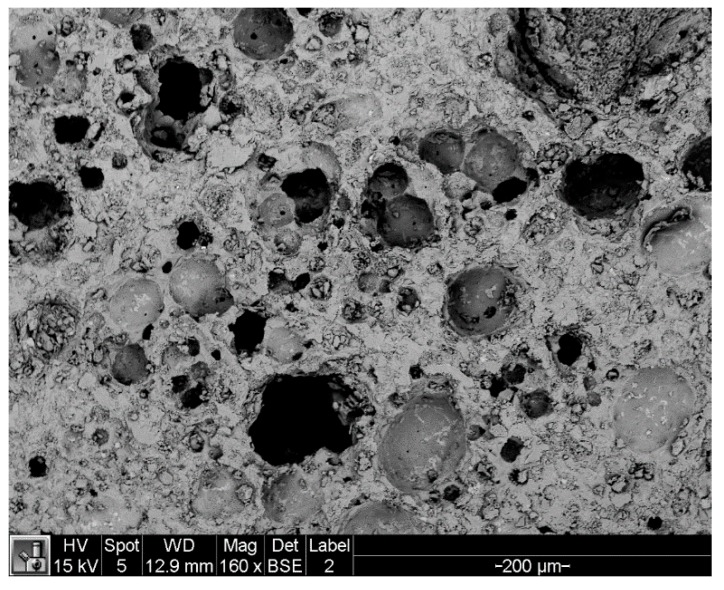
SEM image of LWA in backscattered mode.

**Figure 9 molecules-24-01360-f009:**
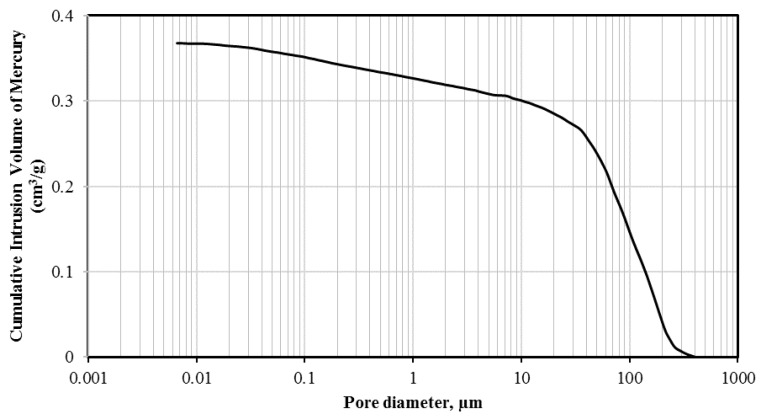
Pore volume of lightweight aggregate measured using MIP.

**Figure 10 molecules-24-01360-f010:**
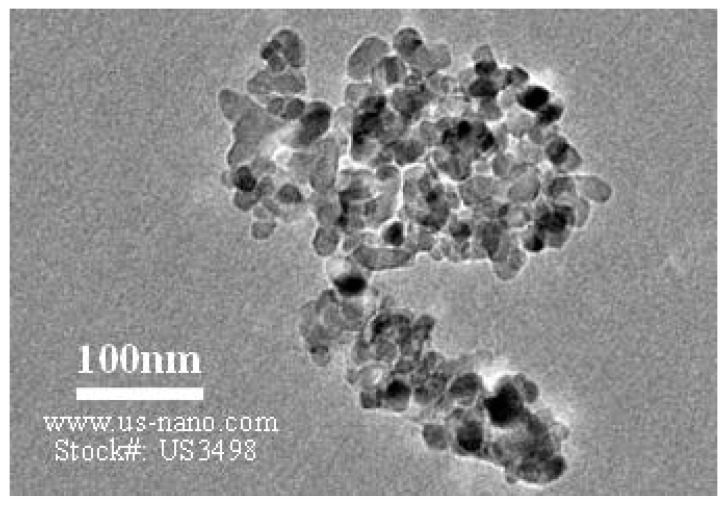
Transmission electron microscopy (TEM) image of nano-titanium.

**Figure 11 molecules-24-01360-f011:**
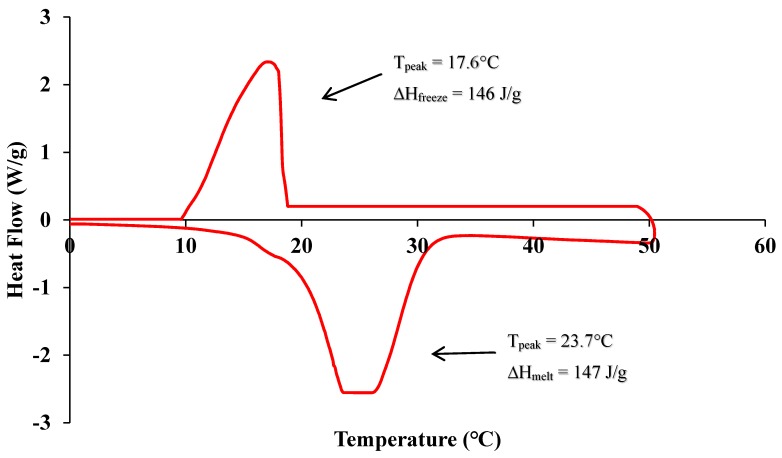
Differential scanning calorimeter (DSC) graph of phase change materials (PCM) used in this study.

**Figure 12 molecules-24-01360-f012:**
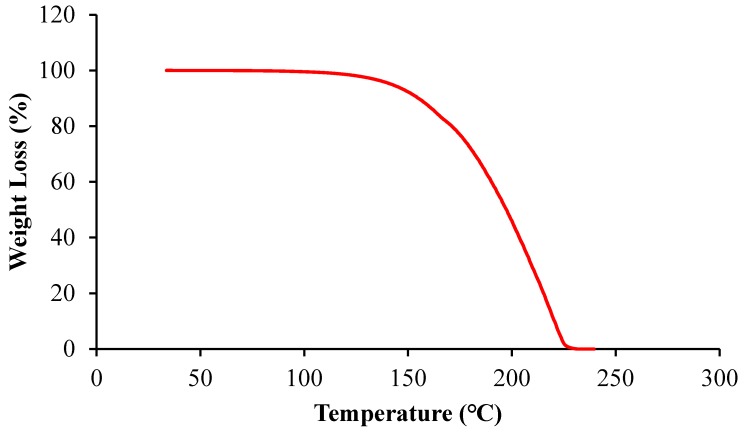
Thermogravimetry analysis (TGA) graph of PCM used in this study.

**Figure 13 molecules-24-01360-f013:**
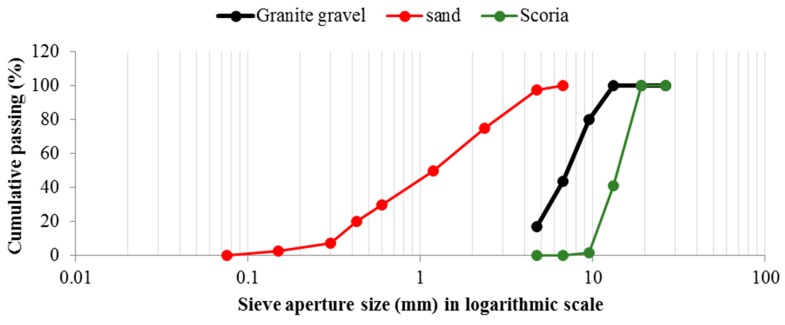
The size distribution of the aggregates.

**Figure 14 molecules-24-01360-f014:**
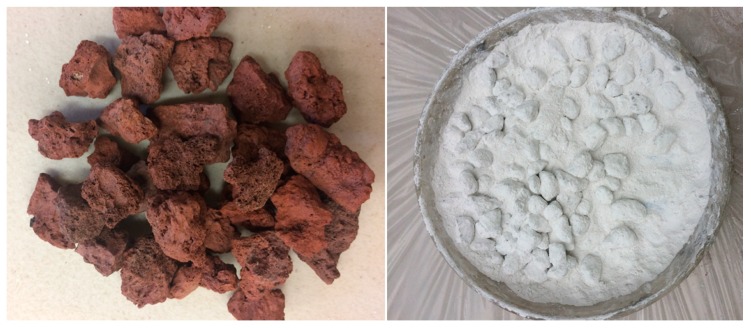
Scoria before and after coating.

**Figure 15 molecules-24-01360-f015:**
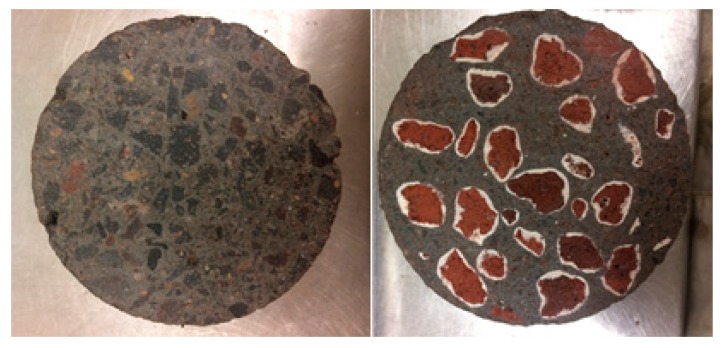
Cross section of control and TESA samples.

**Figure 16 molecules-24-01360-f016:**
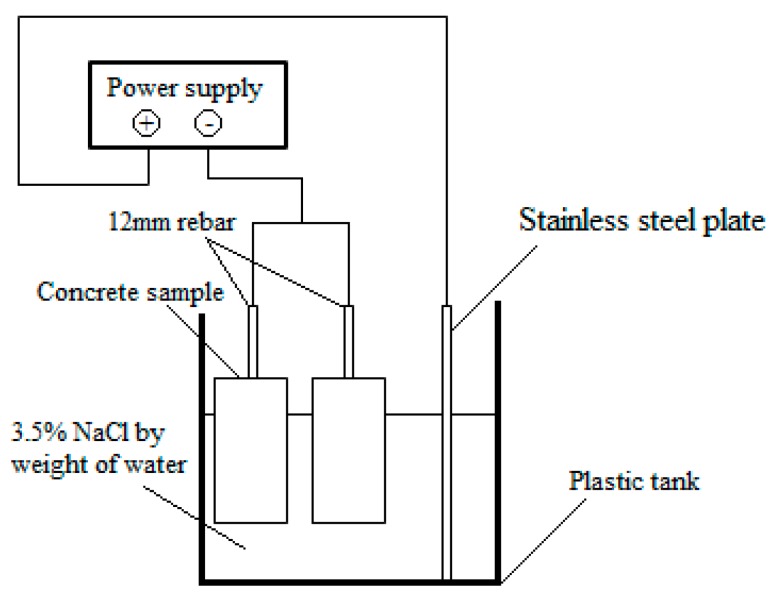
Schematic of the accelerated corrosion test setup.

**Figure 17 molecules-24-01360-f017:**
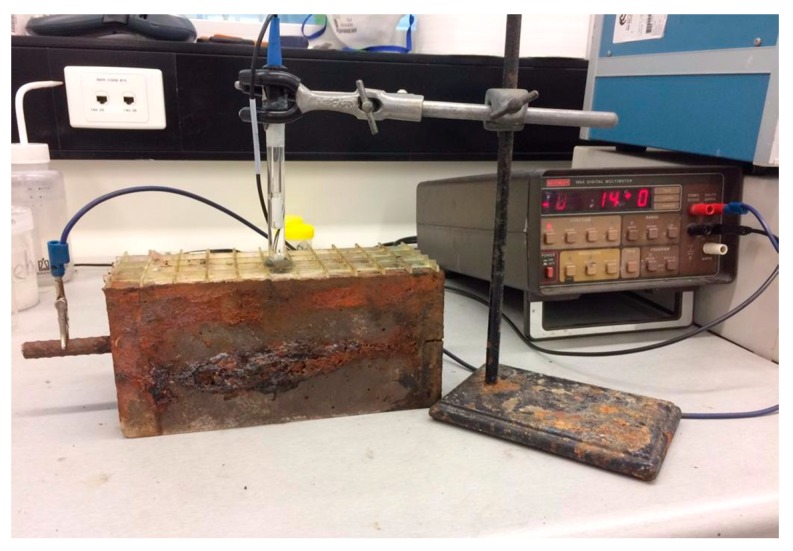
Half-cell potential corrosion measurement.

**Figure 18 molecules-24-01360-f018:**
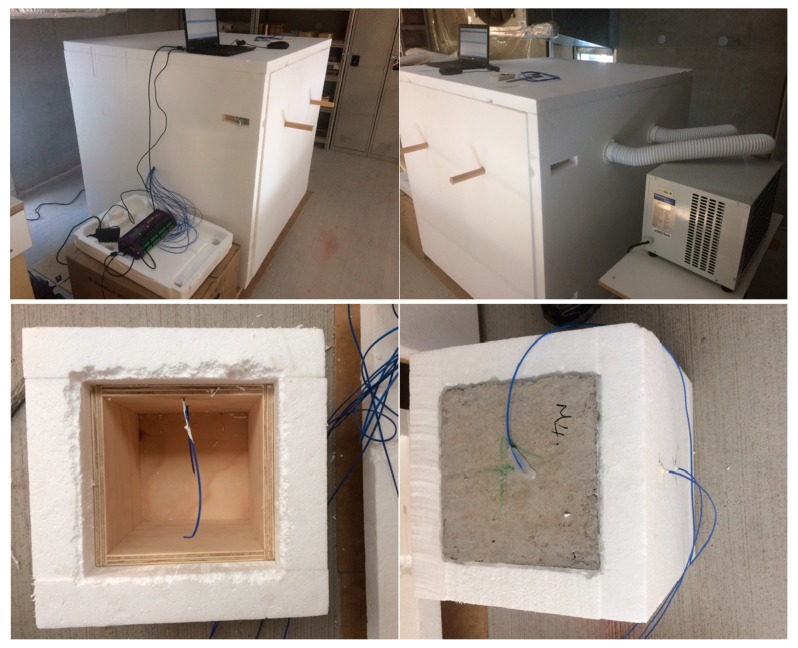
Thermal performance test setup.

**Table 1 molecules-24-01360-t001:** Potential corrosion in reference to half-cell potential ASTM C876-09.

Corrected Half-Cell Potential (mV)	Potential for Corrosion (%)
>−200	10
(−200)–(−350)	50
<−350	90

**Table 2 molecules-24-01360-t002:** MIP test results of scoria as lightweight aggregate.

Aggregate	Volume of Cumulative Mercury Intruded (cm^3^/g)						
		Pore Proportions (%) Classified by Diameter (μm)					
		<0.01	0.01–0.1	0.1–1	1–10	10–100	>100
Scoria	0.3676	0.1	5	7	7	44	37

**Table 3 molecules-24-01360-t003:** Mix proportion of samples (kg/m^3^).

Mix Code	Cement	Fly Ash	Water	Sand	Coarse Aggregate	TESA	NT	HRWR
Control	300	70	140	780	1080	0	0	5
TESA	300	70	140	780	0	692	0	5
TESA-NT	280	70	140	780	0	692	20	7

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
