# Peer review of "Investigation of the Role of Nano-Titanium on Corrosion and Thermal Performance of Structural Concrete with Macro-Encapsulated PCM"

_molecules, 2019, doi:10.3390/molecules24071360_

Round 1

Reviewer 1 Report

The manuscript deals with a very interesting and important problem of searching for structural concrete components in terms of thermal energy storage.

Authors in concrete mix replaced thick aggregate by TESA at 100% by volume of aggregate and NT was added at 5% by weight of cement.

Experimental studies have shown that the compressive strength of 'new' concrete has decreased, while the results of corrosion tests of steel reinforcement in this 'new' concrete are debatable.

The application of the corrosion potential method to assess the corrosion risk of reinforcement is questionable. This method is used in corrosion testing of large concrete structures, but usually at least in combination with concrete resistivity measurements.  It is worth stressing that the results of measurements with the potential method are evaluated in the context of the probability of corrosion of the reinforcement. It should be added that the potential method is typically used for measurements on building structures, while in corrosive laboratory tests polarization measurements should be used. The most popular and effective methods of testing corrosion of reinforcing steel in concrete specimens are linear polarization resistance (LPR) and electrochemical impedance spectroscopy (EIS). Analysis of the results by these methods enables the evaluation of the corrosion rate, which can be expressed in millimetres per year.

In the state of knowledge at the beginning of the manuscript, there is also a clear lack of reference to the very rich literature on polarization tests of the corrosion rate of steel in concrete, which needs to be supplemented.

To sum up, the manuscript is very interesting, but with an incorrectly selected method of corrosion potential to assess the corrosion risk of reinforcement in concrete.

Author Response

Investigation of Role of Nano Titanium in Corrosion and Thermal Conductivity of Structural Concrete with Macro-encapsulated PCM

Ms. Ref. No.: molecules-466902

Responses to reviewer’s comments

The authors would like to thank the editor and the reviewers for their time and useful feedbacks. The manuscript has been revised carefully according to the reviewers’ comments. Please see the detail response to the reviewers indicated below:

Reviewer #1:

The manuscript deals with a very interesting and important problem of searching for structural concrete components in terms of thermal energy storage.

Authors in concrete mix replaced thick aggregate by TESA at 100% by volume of aggregate and NT was added at 5% by weight of cement.

Experimental studies have shown that the compressive strength of 'new' concrete has decreased, while the results of corrosion tests of steel reinforcement in this 'new' concrete are debatable.

1.      The application of the corrosion potential method to assess the corrosion risk of reinforcement is questionable. This method is used in corrosion testing of large concrete structures, but usually at least in combination with concrete resistivity measurements.  It is worth stressing that the results of measurements with the potential method are evaluated in the context of the probability of corrosion of the reinforcement. It should be added that the potential method is typically used for measurements on building structures, while in corrosive laboratory tests polarization measurements should be used. The most popular and effective methods of testing corrosion of reinforcing steel in concrete specimens are linear polarization resistance (LPR) and electrochemical impedance spectroscopy (EIS). Analysis of the results by these methods enables the evaluation of the corrosion rate, which can be expressed in millimetres per year.

The authors agree that the most popular and effective methods of testing corrosion of reinforcing steel in concrete specimens are linear polarization resistance (LPR) and electrochemical impedance spectroscopy (EIS). However, half-cell potential method in accordance with ASTM C876 has been commonly used among researches to quantify the high impedance potentials in concrete specimens, which point to probability of corrosion.  ASTM C876-15 states that the test method is suitable for laboratory concrete research, so the authors used this test method to measure the corrosion potential of composites. Some references have been added in the revised manuscript to justify the use of ACT and half-cell potential test methods for measuring the corrosion of concretes. Please see Lines 179– 187 for details.

2.      In the state of knowledge at the beginning of the manuscript, there is also a clear lack of reference to the very rich literature on polarization tests of the corrosion rate of steel in concrete, which needs to be supplemented.

Thank you for your comment. The linear polarization resistance (LPR) and electrochemical impedance spectroscopy methods have been added in the revised manuscript. Please see Lines 179-180 for details.

To sum up, the manuscript is very interesting, but with an incorrectly selected method of corrosion potential to assess the corrosion risk of reinforcement in concrete.

As mentioned earlier, some references have been added in the revised manuscript to justify the use of ACT and half-cell potential test methods for measuring the corrosion of concretes. Please see Lines 179– 187 for details.

Reviewer 2 Report

The topic is important and up-to-date, as the use of  both nanomodifiers and PCM's in building cement composites are promising and widely investigated in many research and industrial institution. The experimental design and conclusions are suitable in my opinion and can be valuable for the interested readers. I see, however, the necessity of intensive editing job on the text. The English language used is not clear in many places and definitely needs improvement. First of all, the Authors should distinguish the tests from the properties. You can see what I mean already in the title: "... Role of Nano Titanium in Test and Conductivity". Testing is a kind of activity, while conductivity is a property. Does the NT really affect the test? Or the corrosion resistance? The same issue can be found in the text; this needs correction. After this, the paper in my opinion can be published in the Jounal.

Author Response

Investigation of Role of Nano Titanium in Corrosion and Thermal Conductivity of Structural Concrete with Macro-encapsulated PCM

Ms. Ref. No.: molecules-466902

Responses to reviewer’s comments

The authors would like to thank the editor and the reviewers for their time and useful feedbacks. The manuscript has been revised carefully according to the reviewers’ comments. Please see the detail response to the reviewers indicated below:

Reviewer #2:

The topic is important and up-to-date, as the use of both nanomodifiers and PCM's in building cement composites are promising and widely investigated in many research and industrial institution. The experimental design and conclusions are suitable in my opinion and can be valuable for the interested readers. I see, however, the necessity of intensive editing job on the text. The English language used is not clear in many places and definitely needs improvement. First of all, the Authors should distinguish the tests from the properties. You can see what I mean already in the title: "... Role of Nano Titanium in Test and Conductivity". Testing is a kind of activity, while conductivity is a property. Does the NT really affect the test? Or the corrosion resistance? The same issue can be found in the text; this needs correction. After this, the paper in my opinion can be published in the Jounal.

Thank you for dedicating your time to review this paper and proposing the paper for publication in the journal. The manuscript has been carefully revised by a native English speaker to improve the grammar and readability.

According to the comment made by the reviewer, the title has been revised.

Nano titanium particles can not only improve the microstructure of composites by filling the pores, but they can also enhance the thermal conductivity of composite due to their relatively high thermal conductivity. The results can be seen in changes in thermal performance of samples without and with NT.

Reviewer 3 Report

This study presents an experimental study about corrosion test and thermal performance of concrete containing phase change materials and Nano Tio2.

The following sections should be revised for improvement:

1.    Please give a detailed description of the phase change material used in your study.  The relation of phase change process and temperature should be provided.

2.    The additional test about thermal conductivity test of concrete should be added. In your study, you only measured the temperature of concrete. 

3.    How to confirm the binding capacity between LWA and binder when LWA contains PCM. After coating of LWA, the binder capacity will be impaired.

4.    The additional test about chloride diffusivity should be added. The results in Figure 7 relates to the chloride diffusion process.

5.    In lines 195 to 198, you measured steel mass loss. I want to know how to make the corrosion test of steel rebar?  did you measure steel mass loss after accelerated corrosion test ?

6.    In line 185 to line 193, you measured half cell potential of steel rebar. I want to the experimental test setup of half cell potential test? Same as that in figure 7? 

7.    I want to know the degree of saturation of specimens of half cell potential test and accelerated corrosion test.

8.    I want to know the crack condition of the concrete specimen during an accelerated corrosion test.

9.    The range of the Y axis of Figure 13 is too narrow. The room temperature can not be fully shown.

10.    Authors state that Tio2 can fill the pores. Please make the additional experiment of MIP to measure the porosity of specimens.

11. light weight aggregate can lower thermal conductivity of concrete. PCM can uptake and release heat. How to separate the effect of LWA from PCM in your study?

12. can  Ti o2  react with cement? Is tio2 a inert filler material?

Author Response

Investigation of Role of Nano Titanium in Corrosion and Thermal Conductivity of Structural Concrete with Macro-encapsulated PCM

Ms. Ref. No.: molecules-466902

Responses to reviewer’s comments

The authors would like to thank the editor and the reviewers for their time and useful feedbacks. The manuscript has been revised carefully according to the reviewers’ comments. Please see the detail response to the reviewers indicated below:

Reviewer #3:

This study presents an experimental study about corrosion test and thermal performance of concrete containing phase change materials and Nano Tio2.

The following sections should be revised for improvement:

1.      Please give a detailed description of the phase change material used in your study.  The relation of phase change process and temperature should be provided.

Two new figures (Figures 4 and 5) have been added to the revised manuscript. Figure 4 shows the DSC results including the melting and freezing temperatures, and latent heats of the PCM. Figures 5 shows the TGA results of the PCM. Please also refer to Lines 106-110 for details of DSC and TGA tests.

2.      The additional test about thermal conductivity test of concrete should be added. In your study, you only measured the temperature of concrete.

In this study, the thermal performance test was conducted to show the impact of nanoparticles on thermal performance of the composite. The title of the paper has been revised to reflect the content and results of the work presented in the paper. We appreciate the comment made by the reviewer, we will carry thermal conductivity test in the near future.

3.      How to confirm the binding capacity between LWA and binder when LWA contains PCM. After coating of LWA, the binder capacity will be impaired.

In this study, the PCM has been encapsulated within the LWA and the surface of LWA has been fully coated by epoxy and covered by silica fume (as indicated in Figures 7 and 8). Figure 16-b also shows that the interfacial transition zone (ITZ) between coating materials and binder is compacted, indicating a good bonding between PCM-LWA and binder.

4.      The additional test about chloride diffusivity should be added. The results in Figure 7 relates to the chloride diffusion process.

Please kindly note that this paper aimed to report the impact of thermal energy storage aggregate and nano titanium on the most concerned properties of concrete, therefore the compressive strength, probability of corrosion of the reinforcement, thermal performance and microstructure properties have been studied. We appreciate the comment made by the reviewer, we will carry chloride diffusivity test in the near future.

5.      In lines 195 to 198, you measured steel mass loss. I want to know how to make the corrosion test of steel rebar?  did you measure steel mass loss after accelerated corrosion test ?

In order to evaluate the total steel mass loss after the ACT test, the samples were carefully broken open and the rebar was removed and cleaned of any corrosion before weighing. The percentage of mass loss of rebar due to corrosion was recorded as a percentage of its original weight. Please see Lines 211-214 for details.

6.      In line 185 to line 193, you measured half cell potential of steel rebar. I want to the experimental test setup of half cell potential test? Same as that in figure 7?

That is right. Figure 9 (formerly Figure 7) shows the schematic of test setup. A new figure (Figure 10) has been added to show the half-cell potential corrosion measurement.

7.      I want to know the degree of saturation of specimens of half cell potential test and accelerated corrosion test.

Corrosion tanks were filled with a solution containing water with 3.5% salt (NaCl) to assist in accelerating the corrosion within the sample. The bottom part of the samples were fully saturated in water containing 3.5% NaCl.

8.      I want to know the crack condition of the concrete specimen during an accelerated corrosion test.

Though the crack widths have not been measured, the specimens showed similar rusting and cracking.  Please see Lines 278-279 for details.

9.      The range of the Y axis of Figure 13 is too narrow. The room temperature can not be fully shown.

Figure 16 (formerly Figure 13) has been revised to show the room temperature.

10.  Authors state that Tio2 can fill the pores. Please make the additional experiment of MIP to measure the porosity of specimens.

We appreciate the MIP test can measure the porosity of specimens, but we have carried SEM and EDS tests which are also considered the practical experiments to give accurate representation on the porosity of paste and therefore the impact of nanoparticles on paste can also be evaluated.

11.  light weight aggregate can lower thermal conductivity of concrete. PCM can uptake and release heat. How to separate the effect of LWA from PCM in your study?

Previous thermal studies [29] on coated LWA and TESA indicated that TESA showed higher thermal performance compared to coated LWA. The present study herein focused on the influence of TESA and NT on thermal performance of concrete. According to the thermal test results, the peak temperature of sample with TESA was less than control sample by about 2℃. Besides, during the cooling process, the release of heat was slower in TESA sample compared to the control sample, which is again due to the phase change of macro-encapsulated PCM in heat release. Therefore, the changes in the thermal performance test results can be attributed to the PCM. Please see Lines 300-302 for details.

12.  can  Ti o2  react with cement? Is tio2 a inert filler material?

More explanation has been added regarding the chemical reaction of NT and cement paste. Please see Lines 344-349.

Round 2

Reviewer 3 Report

This article is revised.